# Objective Measurements Associated with the Preferred Eating Qualities of Fermented Salamis

**DOI:** 10.3390/foods10092003

**Published:** 2021-08-26

**Authors:** Jihan Kim, Scott Knowles, Raise Ahmad, Li Day

**Affiliations:** Smart Foods Innovation Centre of Excellence, AgResearch Ltd., Te Ohu Rangahau Kai, Palmerston North 4422, New Zealand; Scott.knowles@agresearch.co.nz (S.K.); raise.ahmad@agresearch.co.nz (R.A.); li.day@agresearch.co.nz (L.D.)

**Keywords:** salami, volatile compounds, kokumi, objective measures, product quality, consumer perception, calcium sensing receptor

## Abstract

The development of new food products can be expedited by understanding the physicochemical attributes that are most relevant to consumers. Although many objective analyses are possible, not all are a suitable proxy to serve as quality markers associated with sensory preferences. In this work, we selected nine candidate laboratory assays to use on six commercial salamis, which were also eaten and informally described by a consumer discussion group familiar with China-sourced meat products. Several objective measures were strongly related to the flavour perceptions: (i) texture: instrumental texture values, fat release at oral temperature and fat saturation ratios, (ii) aroma: volatile compounds (e.g., alcohols and esters) associated with microbial fermentation and spices (terpenes and sulphur compounds) and (iii) taste: kokumi taste receptor responses. The fat released at oral temperature was associated with unsaturated fatty acids (r = 0.73). However, there was less explanatory worth for associations between sensory perceptions and proximate composition, water activity, pH or L*, a*, b* colourimetry.

## 1. Introduction

Fermentation is an economical method for adding flavour and extending the shelf-life of traditional meat products. Meat fermentation typically initiates with acidification as carbohydrates are transformed into acids due to the growth of lactic acid bacteria (LAB). LAB releases proteolytic and lipolytic enzymes, generating metabolites and peptides that contribute to desirable taste and other sensory attributes of the fermented meat [1]. Post fermentation, the steps of drying and ripening are employed to concentrate flavours, decrease water activity and improve microbiological stability [2]. 

Peptides and amino acids in foods are notable for their contribution to the taste of umami and the sensation of kokumi, the newest flavour enhancer [3]. Kokumi is particularly advantageous in fermented products as it can synergise with umami, sweet and salty tastes to provide a more balanced and thick flavour. Stimulation of the kokumi calcium-sensing receptor (CaSR) in human taste buds is perceived as mouthfeel, continuity and aftertaste in human sensory analysis [4,5]. 

Dry-cured fermented meats are produced in most countries, with the common sausage format typified by salami. Qualities of the products reflect their ingredients, processing conditions and microbial starter strains. Where ingredients are similar, microbial metabolism becomes the key determinant [4,5]. Regional factors such as climate and storage conditions can influence this microbial growth and help to differentiate textural and flavour characteristics of the products [6]. The resulting foods are diverse and cater to varying preferences. 

Consumers’ perception and liking of the appearance, aroma, taste and texture of food reflects in part their cultural context [7]. In China, where there is a long history of producing fermented meat products, the customary patterns of consumption are in transition due to globalisation. Of the hundreds of different products available in the country, Western-style varieties now account for 55% [8]. A recent survey of 500 consumers in China found that flavour, taste and texture are the main factors considered when buying salami and ham products [9]. 

When a food preference is known (e.g., through surveys, quantitative consumer studies or trained sensory panels) it can become a target for product development. Iterative progress towards that goal is easier if there are physicochemical measurements that can serve as reliable proxies for taste testing. These can be successful for product characterisation, batch comparison and other purposes of quality control. However, their application to interpreting consumer sensory perception is more challenging. 

In the current study, we wanted to determine what types of assays might be useful for measuring the taste, texture and aroma attributes of fermented meat products. For targets, we considered the preferences expressed by a small ‘focus group-style’ discussion of consumers in Beijing, China, who tasted six examples of Western-style salami under the direction of a trained moderator. We then sought to interpret the group’s comments in terms of objective measurements of physicochemical, textural and volatile compound characteristics. To investigate the contribution of kokumi tastants to the salami eating experience, we also used a cell-based CaSR assay to measure the binding of compounds present in aqueous extracts of the meats. This provided insight into the consumer group’s perceptions of flavour complexity, mouthfeel and the subtleties of sweet and salty tastes.

## 2. Materials and Methods

### 2.1. Sample Procurement

Six commercial fermented meat products prepared in a salami format and packaged pre-sliced were purchased from markets in Beijing (see Figure 1). All the products were pork-based and utilised starter cultures of well-defined bacteria. Four were characterised as ‘plain’ (Saucisse, Nostrano, Felino, Toscano) and two as ‘spiced’ (Saucisson and Napoli). They are subsequently abbreviated as: Saucisse (Sse), Nostrano (Nos), Felino (Fel), Toscano (Tos), and as Saucisson (Sau) and Napoli (Nap). 

Retail packs of each salami (90–100 g each) were used in China for tasting by the consumer discussion group and additional packs were imported to New Zealand, where they were stored at 4 °C (up to 1 month) prior to analysis. Four packs of each salami were used for assessment. Replication for each analysis was generally 2–6 slices per pack. Information on the products’ ingredients and starter cultures was provided by the manufacturers and is summarised in Table 1. 

### 2.2. Consumer Discussion Group

A group discussion was held to collect consumers’ opinions about the six salamis. Ten professionals (accountants, engineers, teachers) were recruited in Beijing via telephone interviews. Participation criteria included: gender (50:50 male and female), age (20–40 years old) and consumption habits (fermented meat products at least once per fortnight). The tasting session was managed by a trained moderator who presented the products and facilitated conversation about the appearance, texture and flavour of the samples. Comments by the group were recorded by audio, video and note-taking then subsequently translated. The descriptions regarding individual products were summarised into each parameter (colour, texture and flavour perceptions) and discussed qualitatively, primarily in terms of comparing the rank-orders of the salamis’ descriptions and measurements. 

### 2.3. Proximate Composition, pH and Water Activity

For each product, three slices from a single pack were selected and minced together. Moisture content, total fat and protein as total nitrogen were measured according to AOAC methods by a commercial service (Massey Nutrition Laboratory, Palmerston North, New Zealand). The pH of a homogenate of 2 g sample in 18 mL distilled water was measured with a pH meter (Mettler Toledo, Schwarzenbach, Switzerland). Water activity aw was measured using a Decagon AquaLab 4TE benchtop meter (Pullman, WA, USA). 

### 2.4. Colour

Each pack was opened and then three slices per product were exposed to air for 30 min at room temperature of approximately 25 °C. Colour (L*, lightness; a*, redness and b*, yellowness) of each slice was measured in duplicate using Minolta CR-300 colour meter under Illuminant C lighting (Konica Minolta Photo Imaging Inc., Mahwah, NJ, USA). 

### 2.5. Texture Profile Analysis (TPA)

Sufficient slices of each salami were stacked to 1 cm thickness then trimmed into cuboids (1.0 × 1.5 × 1.5 cm). In order to squeeze out the air between the slices and nominally ‘re-form’ the structure of the original solid sausage format, the stacks were pressed, vacuum packaged (Cryovac^®^ A600 barrier bag, Sealed Air^®^, Charlotte, NC, USA) and refrigerated overnight, then brought to room temperature. Texture characteristics: hardness (unit: kg, the peak force during the first compression cycle), chewiness (unit: kg, the product of hardness, cohesiveness and springiness), cohesiveness (the ratio of the positive force area during the second compression portion to the positive force area during the first compression), springiness (the height that the food recovers during the time that elapses between the end of the first bite and the start of the second bit) were measured using a texture analyser (Stable Micro Systems TA, Surry, UK) with a 50 mm cylinder probe (P/50). The speed was 1 mm/s with a strain of 50% of the original cube height for the samples with a 5 s interval between compression cycles. 

### 2.6. Fat Released

Release of fat was measured by the modified method of Zhao, et al. [10]. Conditions were chosen to roughly approximate mastication. Samples were minced by knife then 2 g placed on folded Whatman filter paper in a centrifuge tube. The tubes were warmed for 15 min in 37 °C water bath to simulate oral temperature. The samples were immediately centrifuged at 2000× *g* for 10 min (Eppendorf centrifuge 5810R, Hamburg, Germany). The total fat and moisture released from the samples was the paper weight difference before and after centrifugation. The moisture of the paper was subsequently evaporated in a drying oven at 105 °C for overnight. The released fat was calculated as the difference between total fluid released and moisture released, expressed as a percentage of sample weight. 

### 2.7. Fatty Acid Composition

Lipids were extracted according to the method of Folch, et al. [11]. Two grams of sample was homogenised with 40 mL chloroform:methanol (2:1, *v*/*v*), filtered through glass wool, then mixed thoroughly with 8 mL of 0.9% sodium chloride solution. The samples were centrifuged at 1000× *g* for 10 min. The upper phase was removed by siphoning and the remaining chloroform phase was evaporated under nitrogen at 55 °C. Methylation reaction was conducted with 0.5 N methanolic sodium hydroxide solution and boron trifluoride solution to generate fatty acid methyl esters (FAME). The fatty acid composition was analysed using Shimadzu GC-2030 (Shimadzu Corporation, Kyoto, Japan) with a flame ionisation detector (FID). The column was a Restek RTX 2330 of 105 m length, 0.25 mm i.d., and 0.20 um film thickness (Restek Corporation, Bellefonte, PA, USA). The thermal programme initial phase was at 70 °C for 2.5 min, increased to 150 °C at the rate of 50 °C/min, then held for 20 min at this temperature; the temperature was then increased to 190 at 25 °C/min and held for 15 min. The injector temperature and detector temperatures were 260 and 265 °C, respectively. Thirty to forty individual FAME were separated and identified by comparing their retention times with those of authenticated standards (Supelco 37 component FAME Mix). Data were expressed in percent of total methyl esters (%). For conciseness, only the summary statistics of fatty acid classes (saturates, polyunsaturates, etc.) are presented. 

### 2.8. Volatile Compounds 

Volatile compounds released from warm salami samples were collected by solid phase microextraction (SPME) and quantified by gas chromatography-mass spectrometry (GC-MS) analysis. Four g of minced sample was placed in a 20 mL glass vial, capped with a laminated Teflon-rubber disk then the vial warmed in a thermal block at 30 °C for 1 h to equilibrate volatiles into the headspace. An SPME fibre was exposed to the headspace at 30 °C for 1 h. Subsequently, the analyses were performed using an GC/MS-QP2010 system (Shimadzu). Analytes were separated using a HP-5 capillary column (Agilent Technologies, 30 m × 0.25 mm and thickness 1 µm), operating at 1 mL/min helium flow. Injection port temperature was maintained at 250 °C and desorption time was 3 min. The temperature program was 50 °C during 3 min, raised to 120 °C at a rate of 5 °C/min, holding for 1 min and then raised to 250 °C at a rate of 8 °C/min, holding for 2 min. The transfer line to the mass spectrometer was held at 240 °C. The mass spectra were obtained by electronic impact at 70 eV. Authenticated standards of n-alkanes (hydrocarbons containing a homologous series, C_n_H_2n+2_) were analysed under the same conditions to calculate the retention indices (RI) for the volatiles. Peaks were identified by comparisons with mass spectra and RI in the library of the National Institute of Standards and Technology (NIST). The likely flavour and aroma characteristics of the detected compounds were assigned based on information in the databases of The Good Scents Company (accessed on-line 2020). To visualise the variation in composition of volatile compounds among different samples, a heatmap was constructed by R 4.0.4 software. Z-scores of concentrations were used to standardise the data and minimise distortions caused by widely different compound levels. 

### 2.9. Kokumi Taste Receptor Response

Kokumi receptor responses were measured using the soluble aqueous fraction of salami according to the method published elsewhere [12,13]. The day before experiment, cells were seeded into 96-well plate and incubated in the incubator for overnight. Next morning, cells were preloaded with 100 µL of fluorescent calcium dye (FLIPR-calcium-6 assay kit, Molecular Devices, San Jose, CA, USA), incubated for 60 min at 37 °C then the salami extract was added to stimulate ligand binding. The assay buffer contained 146 mM NaCl, 5 mM KCl, 0.5 mM CaCl_2_, 2 mM HEPES, pH 7.4. Activation of the CaSR leads to an increase in intercellular calcium ions. The dye binds with high affinity to this free Ca^2+^ and fluoresces at 525 nm when excited at 485 nm. Measurements were made using a fluorescent imaging plate reader (FLIPR; FlexStation3, Molecular Devices, San Jose, CA, USA) and its associated software. The Ca^2+^ response of cells was taken as relative fluorescent units (RFU = maximum RFU-minimum RFU). The final ΔRFU value was calculated by subtracting the response of the control CHO-K1 cells from CHO-K1-CaSR expressing cells. The concentration dependence of fluorescence intensity was evaluated with different concentrations of salami extracts. Nonlinear regression analysis and logistic curves were created with PRISM software version 9 (GraphPad Software Inc., San Diego, CA, USA). Each data point represents the mean ± standard error of the mean (SEM).

#### 2.9.1. Preparation of Aqueous Fraction

Salami samples were homogenised in distilled water with an IKA Ultra Turrax^®^ and centrifuged at 3000× *g* for 15 min at 40 °C. The supernatant was centrifuged at 10,000× *g* for 60 min at 40 °C through a 10 kDa MWCO filter to collect the soluble aqueous fraction containing free amino acids, small proteins and peptides smaller than approximately 70 residues. Total protein content was quantified using BCA protein assay kit (Thermo Fisher Scientific, Waltham, MA, USA) then the material stored at −80 °C until further analysis.

#### 2.9.2. Cell Culture and Media

Cell lines of CHO-K1-CaSR (Chinese hamster ovary cells stably expressing recombinant human CaSR) and CHO-K1 (as a negative control) were purchased (#M00434, GenScript, Piscataway, NJ, USA) and cultured in Ham’s F12, Glutamax media (#31765035, Gibco, Amarillo, TX, USA) supplemented with 10% FBS, 1% penicillin, streptomycin, and 200 μg/mL Zeocin. The cells were grown at 37 °C, 5% CO2. Approximately 70,000 cells were seeded in 100 µL to each well of a 96-well plate (#CLS3603, black/clear flat bottom, Corning, NY, USA). 

### 2.10. Statistical Analysis

The physicochemical properties, texture characteristics, fatty acid and volatile compound compositions of the six salamis were evaluated by one-way analysis of variance (ANOVA, Minitab 18.0). For all analyses except volatile compounds, 6 replicates (*n* = 6) of each sample were assessed. For the volatile compounds, four replicates (*n* = 4) of each sample were assessed. Tukey’s test was used at the 5% level to identify differences among the means of results. Pearson correlation coefficients were performed between fat release and fat content or fatty acid composition. Note that comparisons of laboratory assay results and the consumer discussion group comments are qualitative assessments only. Statistics could not be applied because of non-numerical expression of the group descriptions, which is limitation of this study. 

## 3. Results and Discussion

### 3.1. Consumer Group Assessment

The group of Chinese consumers described their preferences for the colour, texture, flavour and overall liking of salamis, and their consensus was captured. Their observations about eating qualities were sorted by product type and by four categories of sensory perception (visual, texture, aroma and taste) as shown in Table 2. It is acknowledged that the eating experience of salami may be confounded by inhomogeneity inherent in pieces and bites, due to the distribution of particles of fat and meat [14]. 

### 3.2. Product Physicochemical Attributes

In this work, three assays including proximate composition, water activity and pH were considered for sensory characteristics as well as nutritional information about the products. Six additional assays were selected to discriminate product qualities based on each sensory perception (visual, instrumental colour; texture, instrumental texture and fat released as well as fatty acid composition; flavour, volatile compounds and kokumi receptor activation).

### 3.3. Protein, Fat, Moisture and Water Activity 

The moisture content of the six products ranged from 34.1% to 21.9%, ordered as Nos = Nap ≥ Sse = Fel ≥ Tos = Sau (*p* < 0.05, Table 3). Water activity (aw) ranging from 0.89 to 0.84 was correlated with moisture content (Pearson r = 0.85, *p*-value = 0.034). Total protein content ranged from 33.4 to 26.2%, with Fel having the highest and Nos the lowest values. Concomitantly, Fel had the lowest total fat content of 26.6%, whereas Tos had the highest at 36.4%. The proximate composition and water activity are often influenced by the ratio of meat and fat used in the formulation as well as the processing conditions (e.g., the level of dryness) used by different manufacturers. 

### 3.4. Measurements Related to Visual Perception

#### Instrumental Colour Measurement

The colour of meat is often a consumer’s first impression that influences acceptability. In processed meat products, curing colour (pinkish colour) is typically a consequence of nitrosyl-myoglobin derived from nitrite/nitrate or from colourant substitutes, and this positively contributes to consumer preference [15]. The addition of spices to meat products can also influence colour. The two spiced products Sau and Nap showed significantly higher redness (19.8 and 17.0) and yellowness (24.1 and 21.0) compared to the ‘plain’ products that ranged 7.7–11.4 in redness and 9.1–11.9 in yellowness (Table 3). Spicing did not consistently affect lightness: Sau had the lowest value of lightness (39.5), but Nap was intermediate (44.8). 

According to Corcoran, Bernués, Manrique, Pacchioli, Baines and Boutonnet [7], colour preferences can vary depending on culture and country; for instance, the UK prefers meat to be red but not bright, Spain (red, but not dark) and Italy (intense red, not dark). Preferences in China have not been reported. Our small consumer group was put off by the dark, strong colour of Sau. Conversely, they were dissatisfied with plain Tos as being too pale, even though its measured lightness of 44.7 was lower than Sse, which had the highest lightness (50.5) of the six products. Thus, too dark or light colour formation in salamis might be a marker for reduced acceptance. While this measurement technique was able to distinguish among the salamis, it seemed only weakly predictive of the group’s opinions. The limitation of instrumental colourimetry to interpret human-perceived colour of sliced salamis has been attributed to disproportion of lean meat and fat particles [16]. Our eyes may emphasise the predominant colour of the meat part of a salami, whereas a L*, a*, b* colour meter must capture and integrate both its colour and that of the fat. 

### 3.5. Measurements Related to ‘Mouthfeel’ Sensations

#### 3.5.1. Texture Profile Analysis (TPA)

Instrumental texture profile analyses of food based on mechanical double-compression and deformation can be used to imitate the act of biting. In general, main TPA parameters for fermented meat products are hardness, chewiness, springiness and cohesiveness [17]. In particular, hardness (force required to penetrate the sample) and chewiness (energy needed to chew a solid) are important sensory characteristics of salami with respect to satisfaction of consumers [17,18]. 

Our texture analysis revealed significant differences among the six salamis (Table 4). Sse had the highest hardness (17.5 kg) and chewiness (7.06 kg), followed by Tos (14.8 and 5.96 kg) and finally Nap (10.3 and 4.99 kg). The values were ordered as (*p* < 0.001): Hardness (Sse > Tos > Nos > Fel > Sau = Nap), chewiness (Sse > Nos > Tos > Nap > Sau > Fel), Springiness (Tos > Nos > Nap > Fel = Sau> Sse) and Cohesiveness (Nos = Sau = Nap > Sse > Fel > Tos). 

The consumers preferred the texture of the Tos product followed by Nos, which had a moderate level of chewiness according to instrumental measurement. In contrast, Sse was too chewy and firm textured, which was reflected in high instrument scores on chewiness and hardness. It appears that the participants preferred salami products with moderate hardness (13–15 kg) and chewiness (5.9–6.5 kg). 

#### 3.5.2. Fat Release 

The amount of fat released during mastication contributes to mouthfeel and enhances palatability of foods [19]. The percentage of fat released from the six salamis is shown in Table 4. There was greater than 5-fold difference between products (*p* < 0.001). Nos and Nap released the most fat, at 4.5–5.0%. Sse and Sau released less than 1%, with Fel and Tos in the intermediate range of 1.8–2.2% fat. Fat release was not correlated to the samples’ total proximate fat content (r = 0.24). 

Camacho, et al. [20] demonstrated that the amount of fat released from a food matrix contributes to intensity of the mouthfeel sensation. Likewise, a positive contribution of melted fat to mouthfeel was reported, whereas incompletely melted fat in the mouth provides undesirable feeling [19]. This seems to be the case with the discussion group’s observations. These consumers enjoyed the sensations of greasy, softness and melting in the mouth with Nos, while the greasy of Sse was poorly perceived. The results suggest that measuring the amount of fat released (in contrast to simply total fat content, see above) can be predictive of sensations and preferences associated with mouthfeel. 

### 3.6. Fatty Acid Composition

Fatty acids have a wide range of melting points and this contributes to the texture and mouthfeel perception of fat-rich foods [21]. Generally, a high proportion of solid fats increases the firmness of processed meat products and this leads to better eating quality [22]. 

Summary statistics of fatty acid classes for the six salamis are shown in Table 4. Fel and Tos had marginally more saturated fatty acids (SFA, 38.6% and 39.1%) compared to the other four products (34.2–37.7%) and correspondingly the lowest levels of unsaturated fatty acids (UFA, 60.3% and 60.1%). Maximum UFA were in Nos and Nap (63.2% and 64.6%). The lowest ratios of UFA/SFA were in the Fel and Tos products, whereas the highest ratios were shown in Nos and Nap (1.45–1.53 versus 1.76–1.88). As shown in Table 4, UFA content of the salamis positively correlated with their fat released at 38 °C (r = 0.73), whereas the SFA of the salamis negatively correlated (r = −0.71). This likely reflects the lower melting point of the unsaturates, which is for example <15 °C for oleic and linoleic acids but >65 °C for palmitic and stearic acids [23].

The discussion group described positive mouthfeel from the Nos and Nap salamis in part due to the melting in the mouth, and this sensation may have a strong responsibility for overall texture perception by untrained tasters. The purposeful manipulation of fatty acid composition to achieve different melting points and concomitant mouthfeels could be useful during product development and optimisation.

### 3.7. Measurements Related to Aroma Perception

#### Volatile Compounds

Salami is well known for delivering rich and complex aromas. These are derived from volatile compounds associated with process conditions such as fermentation (e.g., temperature, microbial activity, and ripening time) and additives (e.g., onion, garlic, and pepper) [24,25]. 

Our analysis of volatiles detected 81 compounds in the headspace above warmed samples of salami (Table 5 and Figure 2). These included 10 alcohols, 6 esters, 9 aldehydes, 10 ketones, 4 acids, 4 unspecified hydrocarbons, 28 terpenes, 7 sulphur related compounds and 12 unidentified or contaminants. There was substantial variation among the six products. All the identified volatile compounds are listed in Table 5 and visualised with a heatmap in Figure 2. 

Sse and Sau showed the most alcohol derivates (26.7% and 37.5%, compared to 8.73–19.0%). Particularly, a remarkably high level of ethanol in Sau was detected (Table 5). It is known that the ethanol formation is associated with carbohydrate catabolism manipulated by microorganisms [26]. Other alcohols produced mainly by lipid oxidation were identified as desirable aroma compounds in Chinese Cantonese sausage [27]. 

Sau also showed the highest concentration of volatile esters (6.59%), being 6 to 20-fold greater than the other products. The compounds contributing substantially were methyl and ethyl acetates, which likely produced fruity aromas. Esters of Sau accounted for 6.59%, which was much higher than others (0.36% to 1.05%). The esters are strongly responsible for aroma of fermented foods due to their lower threshold values. According to the group discussion, the Sau was selected as an ideal product possessing a desirable mixture of aromas. The higher esters of Sau could contribute to the positive group perception. The presence of alcohol promotes the ester synthesis in fermented meat [28]. Thus, the abundant ethanol in Sau could contribute to the ester formation providing desirable aroma to consumers. 

Tos and Fel were rich in aldehydes, accounting for 15.6% and 19.4% of total peak area in chromatograms. This was higher than the other four products (2.11–9.16%). Their dominant compound was hexanal. Volatile aldehydes can be formed from oxidised fatty acids, such as decanal (from oleic acid), hexanal (linoleic acid) and propanal (linolenic acid). The aldehydes derived from fatty acid autoxidation are highly associated with the undesirable flavour such as greasiness, metallic as well as rancidity [25]. 

The aroma of Tos was not acceptable to the consumers due to strong oily notes. This product showed the greatest total chromatogram peak area, which suggests it had the greatest overall aroma intensity. Yet the aroma of Fel was described as a good smell despite similar content of hexanal (9.01 vs. 10.6). The greasy character of Fel may have been masked by other flavours. For instance, branched aldehydes transformed from branched amino acids via amino acid catabolism can impart desirable flavour [29]. In this study, 2-methyl propanal, 2- and 3- methyl butanal of Tos and Fel were higher than those of others.

Fel and Tos were also rich in ketones (20.7 and 16.5% versus 5.55–14.8%). Ketones are generally formed either via decarboxylation or β-oxidation [30]. The main ketone was 2-butanone, likely a buttery aroma, whereas its 3-hydroxy-derivative was the major compound in Sau. A 3-hydroxy-2-butaone responsible for desirable flavour is produced via microbial fermentation [31]. 

The terpenes and sulphur compounds likely originated from the mixture of spices and peppers, particularly the garlic and onion that belong to Allium species. Sse and Nap had a higher level of terpenes than other products (41.7% and 42.6% versus 6.03–29.6%). The Nos showed the highest level of sulphur compounds (19.9%) followed by Tos, Sau and Nap (12.1%, 10.3% and 11.0%). The discussion group positively described the intensive garlic flavour with strong meat aroma of Sau possessing the secondary highest level of ally methyl sulphide which is known as a garlic derivate. 

It has been reported that hexanal derived from lipid autoxidation contributes to the highest consumer acceptance of Western type-high fat fermented sausage [32]. On the other hand, ethanol and alcohol esters are the main volatile compounds that define flavour characteristics of Chinese dry-cured sausage whereas the hexanal was not a major contributor in perspective of Chinese consumer [24]. Taken together, the fermentation derivatives from microbial reactions with sulphur-containing volatile compounds derived from Allium species can provide familiarity and desirable notes that enhance Chinese consumer preference more rather than foreign notes derived from lipid autoxidation.

### 3.8. Measurements Related to Taste Perception

#### 3.8.1. pH and Sourness

The pH of fermented foods is primarily a consequence of microbial metabolism driven by choice of starter strain and availability of carbohydrates. The accumulation of organic acids (e.g., lactic acid and acetic acid) by glycolysis reduces pH, whereas the release of nitrogenous compounds (e.g., ammonia, peptides and free amino acids) as a result of proteolysis tends to neutralise the pH of fermented meat products [2]. Marked differences in pH were observed among the six products in this study, ordered from high to low as: Nos (pH = 5.85) ≥ Sse > Nap > Tos = Fel ≥ Sau (pH = 4.8) as shown in Table 2. 

While the level of acidification influences perceived sourness, it is not the only determinant, as different organic acids can differ in potency at the same pH [33]. The acidic Fel and Tos products were described as having strong sourness, but the consumers did not report strong sourness from Sau despite its very low pH.

#### 3.8.2. Saltiness, Sweetness and Spicy

In addition to the acids of sourness, the taste of fermented meat depends on a range of non-volatile soluble compounds that are not so simply perceived or easily measured. These include protein degradation products (e.g., peptides and free amino acids) and adjuncts (e.g., salt, sweeteners and capsaicin) [5,34]. The discussion group described some taste characters of sweetness and saltiness for the salamis, and also spicy-hot tastes for Sau and Nap (Table 1). However, the comments were vague and varied, as would be expected from untrained tasters. Only Sau received any direct mention about saltiness, as “good salty”. According to the nutrient label information of each product (data not shown), the sodium content of Sau and Nap (spiced types) was 1930 and 1952 mg/100 g meat. This was slightly less than the others, which ranged from 1960 to 2200 mg/100 g meat. In general, the sodium content of a meat product is determined by formulas and manufacturing recipes. 

Moderate sweetness in four of six salamis had a positive impact. Dry-cured meat products possessing sweetness are familiar to Chinese cuisine [35].

#### 3.8.3. Kokumi Taste Receptor Response

The kokumi flavour sensation is imparted by small γ-glutamyl peptides and aromatic amino acids (tryptophan, tyrosine, phenylalanine) that are naturally enriched in some foods such as garlic, onion, mushroom, beans and fermented ingredients [3]. Kokumi synergises with the taste response to glutamate to enhance the intensity of umami, sweetness and saltiness, and the depth of overall flavour, resulting in consumer satisfaction [12,36]. 

Figure 3 shows that all salami samples except Nos and Sse contained measurable amounts of kokumi tastants that activated the kokumi receptor in a concentration-dependent manner. The flavoured products, in particular, Sau caused strong activation and high potency (lowest value for half-maximal effective concentration (EC50 > 0.29 mg) followed by Nap (EC50 > 0.34 mg), Fel and Tos which caused moderate activation. Sse and Nos caused only low level of activation at the highest applied dose. The CaSR responses suggest that Sau is especially rich in kokumi tastants. This likely contributed to the consumer group’s description of Sau as having balanced sweetness, saltiness. They mostly preferred it to other salamis despite it having lower levels of fatty acids than the other product, which might otherwise be thought to confer the desirable mouthful/creamy flavour sensation. Our results are consistent with Ohsu, Amino, Nagasaki, Yamanaka, Takeshita, Hatanaka, Maruyama, Miyamura and Eto [12], for whom the kokumi receptor assay was observed to support sensory perception of enhanced umami, salty and sweet taste perception. However, to elucidate the relationship between kokumi receptor activation assay and human taste perception, a specifically-defined sensory analysis with taste panellists will be required. 

In the current study, we did not directly assess all non-volatile tastants, other than labelled sodium content as a proxy for saltiness, pH as a proxy for sourness and kokumi taste receptor for overall taste intensity. We assume that interaction among tastants could have contributed to consumers’ perceptions. For instance, concomitant exposure to a spicy component such as capsaicin is known to enhance salt sensitivity [37]. The strong spiciness added to Sau could mask the other tastes, resulting in the reduction of sourness intensity. The visual and texture perceptions of Sau were moderate, yet it was described as the highest overall preference. This might be influenced by kokumi tastants that created a more balanced and enhanced flavour. 

## 4. Conclusions

In this study, we found that a number of objective measures were useful for interpreting how consumers perceived fermented meat products. After tasting six salamis, the preferences of the discussion group were summarised as (1) colour: not too dark or light; (2) texture: excellent mouthfeel with melted fat in the palate and moderate texture strengths; (3) aroma: aromatic volatile compounds, which are derived from microbial fermentation, as well as culturally-familiar sulphur-containing aromas; (4) taste: moderate levels of sweetness, saltiness and sourness, which was further examined through the kokumi receptor response in terms of synergy between these tastes for enhanced flavour and palatability. Apparent associations between the tasting comments and the measured values were seen for: total fat content; texture hardness and chewiness; fat release at 37 °C (oral temperature); fatty acid UFA/SFA ratio; and volatiles of terpene and sulphur types related to spices. There was less explanatory worth for: total protein, aw, L*, a*, b* colourimetry, pH, and volatiles of the dominant aldehyde types. 

## Figures and Tables

**Figure 1 foods-10-02003-f001:**
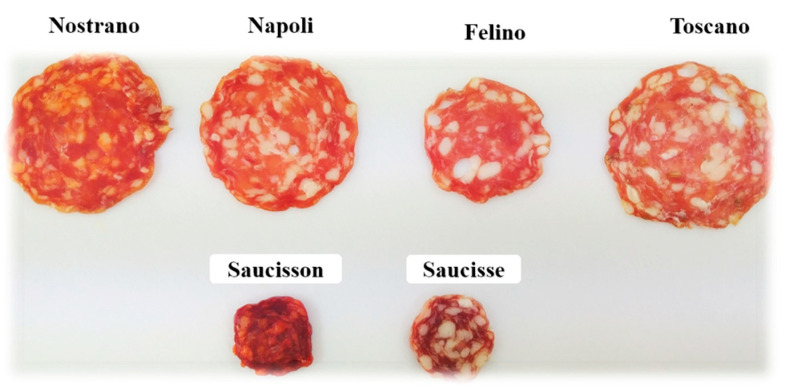
Photos of six examples of pre-sliced commercial fermented salami from China. Nos and Nap are produced by Yuran, Fel and Tos by Casa Modena, and Sse and Sau by Bastides.

**Figure 2 foods-10-02003-f002:**
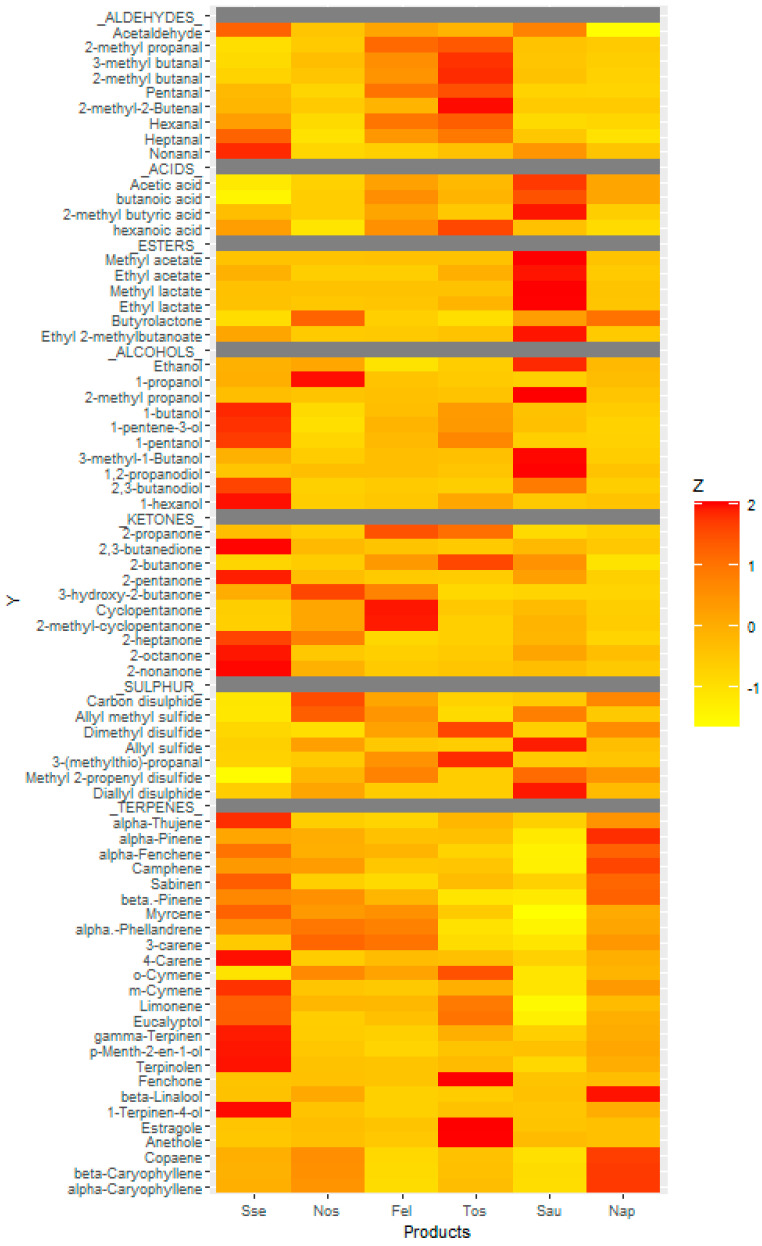
Heatmap visualization based on the volatile compounds of six salamis. z-scores were calculated in order to standardise the data based on the six salamis, and minimise distortions caused by different compound levels. The colour of each tile of the heatmap presents the abundance of each volatile compound among the six salamis. The red colour indicates major abundance, while the yellow colour indicates minor abundance.

**Figure 3 foods-10-02003-f003:**
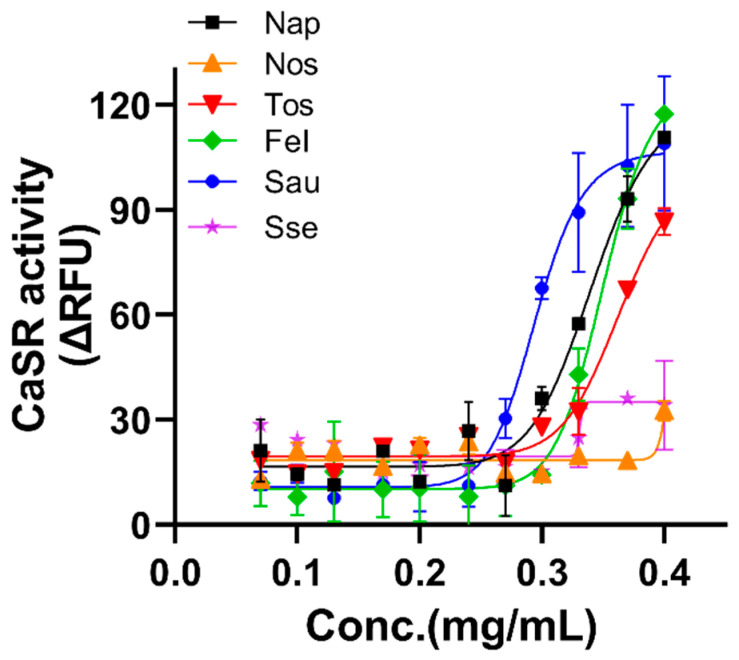
Dose response curve of the kokumi receptor (CaSR) response against water-soluble extracts of six salamis. The values are presented as relative fluorescent units (ΔRFU) which are normalized by subtracting the RFU values of mock cells (CHO-K1) from those expressing CaSR. EC50 (half maximal effective concentration) was calculated by non-linear regression analysis in GraphPadprism 9 software. Values are mean ± SEM, *n* = 3. Nap (EC50 > 0.34 mg), Nos (EC50 > 0.4 mg), Tos (EC50 > 0.36 mg), Fel (EC50 > 0.35 mg), Sau (EC50 > 0.29 mg), Sse (EC50 > 0.3 mg).

**Table 1 foods-10-02003-t001:** Ingredient information of six commercial fermented salami from China.

	Plain Types				Spiced Types	
	Sse	Nos	Fel	Tos	Sau	Nap
Ingredients	pork, salt, spices, dextrose, sugar, lactose, potassium nitrate	pork, salt, glucose, spices, food additives (sodium-D-isoascorbate, sodium nitrite and potassium nitrate)	pork, salt, glucose, sugar, food additives (ascorbic acid and sodium nitrite)	pork, salt, glucose, sugar, food additives (ascorbic acid and sodium nitrite)	pork, salt, spices (red pepper 3.4%), dextrose, glucose syrup sugar, lactose, potassium nitrate	pork, salt, glucose, spices including hot spicy, food additives (sodium-D-isoascorbate, sodium nitrite and potassium nitrate)
Starter cultures	*L. plantarum*	*L. sakei* *S. xylose*	*P. acidilactici* *P. pentosaceus*	*P. acidilactici* *P. pentosaceus*	*L. plantarum*	*L. sakei* *S. xylose*

Products: Sse (Saucisse), Nos (Nostrano), Fel (Felino), Tos (Toscano), Sau (Saucisson) and Nap (Napoli).

**Table 2 foods-10-02003-t002:** Summary of comments from a Chinese focus group regarding six commercial fermented meat products of either plain or spiced type.

	Plain Types				Spiced Types	
	Sse	Nos	Fel	Tos	Sau	Nap
Visual	-	-	-	Negative perception on lighter colour Looks too oily	Negative perception on dark colour	Looks spicy
Texture	Too chewy and firm A little greasy	Loose fat and greasy Soft Melt in the mouth	Chewy Dry and firmChecked Greasy	Good texture Good chewiness, Too oily	Not greasy, Chewy and soft	Good chewiness Moderate softness and firmness
	Ideal chewy ranking: Tos, but the focus group concluded Nos product as ideal texture product					
Aroma	Like it Meat aroma, cheese flavour, No aftertaste or very light.	Dislike Less flavour Less aftertaste	Smell good Less aftertaste	Dislike Strong oily flavour	Like it Garlicy Smoky Hot, mix of different meat flavour	Dislike Spicy flavour overwhelming meat flavour
Taste	Moderate sweet and salt.	Moderate salty, sweet and sour	Strong sour Moderate salty and sweet	Strong sourness Too salty	Sweet, sour and spicy, good salty	Spicy, but does not last long Not sweet
	Ideal flavour ranking: Sau > Nos = Nap > Sse > Fel > Tos					
Overall preference *	4.2 ^ab^	3.8 ^abc^	3.3 ^bc^	2.7 ^c^	4.8 ^a^	4.2 ^ab^
Overall preference ranking: Sau ≥ Nap = Sauc ≥ Nos ≥ Fel ≥ Tos						

* Overall preference was scored with 5-point scale ranging from 1 = very dislike to 5 = very like. Means in the same column followed by different letters are significantly different by Tukey’s test (*p* < 0.05). Products: Sse (Saucisse), Nos (Nostrano), Fel (Felino), Tos (Toscano), Sau (Saucisson) and Nap (Napoli).

**Table 3 foods-10-02003-t003:** Product quality attributes including proximate composition, pH, water activity and instrumental colour (lightness L*, redness a*, yellowness b*) of six commercial fermented meat products (*n* = 6).

	Plain Types				Spiced Types		
	Sse	Nos	Fel	Tos	Sau	Nap	SEM ^a^
Moisture	27.2 ^ab^	34.1 ^a^	25.3 ^ab^	22.3 ^b^	21.9 ^b^	33.2 ^a^	0.57
Protein	27.7 ^b^	26.2 ^b^	33.4 ^a^	29.7 ^ab^	29.1 ^ab^	27.8 ^b^	0.75
Fat	32.2 ^bc^	30.0 ^c^	26.6 ^d^	36.4 ^a^	33.7 ^b^	30.5 ^c^	0.94
pH	5.76 ^a^	5.85 ^a^	5.19 ^b^	5.25 ^b^	4.88 ^c^	5.43 ^bc^	0.06
Aw ^(1)^	0.87 ^bc^	0.89 ^a^	0.84 ^e^	0.84 ^de^	0.86 ^cd^	0.88 ^ab^	0.00
L*	50.4 ^a^	48.4 ^ab^	45.1 ^b^	44.7 ^b^	39.5 ^c^	44.7 ^b^	1.01
a*	7.72 ^d^	11.4 ^cd^	11.3 ^cd^	10.0 ^cd^	19.8 ^a^	17.0 ^b^	0.63
b*	11.8 ^c^	11.1 ^cd^	9.07 ^d^	10.0 ^cd^	24.1 ^a^	20.9 ^b^	0.51

^(1)^ Aw: water activity; standard error of the mean. Means in the same row followed by different letters are significantly different by Tukey’s test (*p* < 0.05). Products: Sse (Saucisse), Nos (Nostrano), Fel (Felino), Tos (Toscano), Sau (Saucisson) and Nap (Napoli).

**Table 4 foods-10-02003-t004:** Product texture attributes including instrumental texture analysis, amount of fat released (%), and fatty acid composition (% of total methyl esters) of six commercial fermented meat products (*n* = 6).

	Plain Types				Spiced Types		
	Sse	Nos	Fel	Tos	Sau	Nap	SEM
*Instrumental texture analysis*							
Hardness (kg)	17.5 ^a^	13.0 ^bc^	11.9 ^bc^	14.8 ^ab^	10.8 ^c^	10.3 ^c^	0.38
Springiness	0.69 ^b^	0.79 ^ab^	0.71 ^ab^	0.83 ^a^	0.71 ^ab^	0.76 ^ab^	0.01
Cohesiveness	0.58 ^b^	0.63 ^a^	0.52 ^c^	0.48 ^c^	0.62 ^a^	0.63 ^a^	0.01
Chewiness (kg)	7.06 ^a^	6.54 ^ab^	4.56 ^b^	5.96 ^ab^	4.73 ^b^	4.99 ^b^	0.19
*Oral processing simulation*							
Fat released (%)	0.96 ^c^	4.5 ^a^	1.8 ^b^	2.2 ^b^	0.86 ^c^	5.0 ^a^	0.12
*Fatty acid composition* (%)							
SFA	37.7 ^ab^	35.7 ^cd^	38.6 ^a^	39.1 ^a^	36.6 ^bc^	34.2 ^d^	0.290
MUFA	48.9 ^a^	45.7 ^b^	42.3 ^c^	42.7 ^c^	48.5 ^a^	45.0 ^b^	0.408
PUFA	12.1 ^c^	17.4 ^b^	18.0 ^ab^	17.3 ^b^	13.7 ^c^	19.5 ^a^	0.408
UFA	61.1 ^cd^	63.2 ^ab^	60.3 ^d^	60.1 ^d^	62.3 ^bc^	64.6 ^a^	0.278
UFA/SFA	1.63 ^cd^	1.76 ^ab^	1.45 ^d^	1.53 ^d^	1.70 ^bc^	1.88 ^a^	0.020

SFA: saturated fatty acid; MUFA: monounsaturated fatty acid; PUFA: polyunsaturated fatty acid; UFA: unsaturated fatty acid; SEM, standard error of the mean. Means in the same column followed by different letters are significantly different by Tukey’s test (*p* < 0.05). Products: Sse (Saucisse), Nos (Nostrano), Fel (Felino), Tos (Toscano), Sau (Saucisson) and Nap (Napoli).

**Table 5 foods-10-02003-t005:** Product aroma attributes of volatile compounds (% of area relative to the total chromatographic peak area of the identified peaks) via solid phase microextraction, gas-chromatography-mass spectrometry (SPME GC-MS) of six fermented meat products (*n* = 4).

Compound				Plain Types				Spiced Types	
	Retention Time	Retention Index	*m*/*z*	Sse	Nos	Fel	Tos	Sau	Nap
*Aldehydes*				9.16	2.85	15.67	19.43	3.69	2.11
Acetaldehyde	1.398	412	44	2.37	1.23	1.64	1.46	2.05	0.47
2-methyl propanal	2.393	535	72	0.06	0.30	1.33	1.47	0.38	0.28
3-methyl butanal	4.002	649	58	0.20	0.79	1.77	3.15	0.63	0.45
2-methyl butanal	4.247	661	86	0.03	0.07	0.20	0.40	0.08	0.04
Pentanal	5.03	698	58	0.56	0.12	1.55	1.95	0.14	0.13
2-methyl-2-Butenal	6.442	748	84	0.04	0.00	0.05	0.26	0.00	0.00
Hexanal	8.31	811	56	5.71	0.33	9.01	10.60	0.34	0.71
Heptanal	11.809	902	70	0.16	0.01	0.11	0.14	0.05	0.01
Nonanal	18.947	1102	82	0.01	0.01	0.01	0.01	0.01	0.01
*Acids*				3.32	5.70	10.65	8.22	18.21	10.16
Acetic acid	3.03	602	60	3.09	5.41	10.02	7.72	17.33	9.71
butanoic acid	7.472	790	60	0.12	0.27	0.48	0.35	0.64	0.41
2-methyl butyric acid	9.662	846	74	0.03	0.01	0.06	0.01	0.19	0.00
hexanoic acid	14.161	977	60	0.08	0.02	0.09	0.13	0.05	0.03
*Esters*				0.91	0.36	0.33	1.05	6.59	0.40
Methyl acetate	2.104	515	74	0.02	0.02	0.05	0.04	1.95	0.02
Ethyl acetate	3.18	612	61	0.79	0.23	0.22	0.82	2.75	0.28
Methyl lactate	6.402	754	45	0.02	0.02	0.02	0.02	0.76	0.01
Ethyl lactate	8.757	821	45	0.05	0.01	0.01	0.15	1.04	0.02
Butyrolactone	12.222	915	86	0.02	0.08	0.03	0.02	0.05	0.07
Ethyl 2-methylbutanoate	10.022	850	102	0.01	0.00	0.00	0.00	0.04	0.00
*Alcohols*				26.73	19.03	8.73	14.39	37.50	14.59
Ethanol	1.605	440	45	14.29	16.03	5.93	9.70	30.35	12.88
1-propanol	2.36	533	59	0.79	2.49	0.43	0.32	0.21	0.54
2-methyl propanol	3.357	639	74	0.03	0.01	0.02	0.01	0.52	0.01
1-butanol	4.159	668	56	0.26	0.01	0.06	0.12	0.06	0.04
1-pentene-3-ol	4.673	686	57	2.85	0.05	0.93	1.42	0.67	0.31
1-pentanol	7.113	766	55	1.48	0.04	0.37	0.90	0.12	0.10
3-methyl-1-Butanol	6.102	730	55	0.78	0.12	0.44	0.41	3.33	0.11
1,2-propanodiol	6.212	732	45	0.03	0.08	0.08	0.03	0.78	0.05
2,3-butanodiol	7.854	800	45	1.72	0.07	0.14	0.10	1.20	0.10
1-hexanol	10.602	869	56	4.50	0.13	0.33	1.35	0.26	0.46
*Ketones*				11.54	14.85	20.75	16.51	8.45	5.55
2-propanone	1.825	503	58	4.98	4.09	11.03	9.69	3.22	3.88
2,3-butanedione	2.747	579	86	0.74	0.14	0.08	0.04	0.14	0.06
2-butanone	2.892	601	72	1.61	1.98	3.92	6.36	4.16	1.04
2-pentanone	4.759	685	58	0.23	0.04	0.02	0.02	0.09	0.01
3-hydroxy-2-butanone	5.438	706	88	2.86	7.86	5.22	0.24	0.44	0.44
Cyclopentanone	7.987	797	55	0.03	0.12	0.34	0.04	0.07	0.03
2-methyl-cyclopentanone	9.748	847	55	0.01	0.03	0.10	0.00	0.03	0.01
2-heptanone	11.4	893	58	0.82	0.55	0.04	0.09	0.24	0.05
2-octanone	14.97	993	58	0.15	0.01	0.00	0.01	0.05	0.02
2-nonanone	18.499	1093	58	0.13	0.03	0.00	0.01	0.01	0.00
*Sulphur compounds*				1.46	19.99	12.18	4.43	11.09	10.37
Carbon disulphide	2.271	522	534	1.40	10.61	5.81	2.81	3.32	7.63
Allyl methyl sulfide	5.078	700	701	0.00	9.18	5.89	0.95	7.32	2.23
Dimethyl disulfide	6.588	758	761	0.06	0.03	0.25	0.52	0.09	0.33
Allyl sulfide	10.421	868	862	0.00	0.04	0.01	0.00	0.11	0.01
3-(methylthio)-propanal	11.995	907	906	0.00	0.00	0.03	0.07	0.00	0.01
Methyl 2-propenyl disulfide	12.519	922	919	0.00	0.11	0.18	0.08	0.21	0.16
Diallyl disulphide	18.325	1088	1082	0.00	0.01	0.00	0.00	0.04	0.01
*Terpenes*				41.73	29.69	24.61	27.45	6.03	42.67
alpha-Thujene	12.908	925	93	4.63	0.35	0.12	1.13	0.26	2.28
alpha-Pinene	13.221	939	93	5.35	5.00	3.66	3.74	1.01	11.01
alpha-Fenchene	13.709	951	93	0.03	0.02	0.02	0.01	0.01	0.04
Camphene	13.805	953	121	0.13	0.13	0.07	0.08	0.02	0.21
Sabinen	14.586	977	93	3.85	0.46	0.06	1.07	0.37	3.65
beta-Pinene	14.804	981	93	6.48	6.09	4.10	1.48	1.27	8.18
Myrcene	15.031	992	69	0.48	0.35	0.37	0.21	0.05	0.31
alpha.-Phellandrene	15.667	1006	93	0.50	0.60	0.56	0.14	0.05	0.42
3-carene	15.895	1012	93	3.26	10.12	9.36	1.93	1.28	7.09
4-Carene	16.08	1022	121	0.67	0.03	0.11	0.09	0.01	0.18
o-Cymene	16.223	1020	119	0.05	0.28	0.22	0.39	0.04	0.17
m-Cymene	16.348	1026	119	6.18	1.72	1.56	2.56	0.51	3.37
Limonene	16.529	1031	68	5.11	2.70	2.67	4.48	0.57	2.61
Eucalyptol	16.622	1046	93	3.70	1.30	1.63	3.32	0.45	2.05
gamma-Terpinen	17.504	1064	93	0.80	0.04	0.01	0.22	0.02	0.23
p-Menth-2-en-1-ol	17.83	1145	93	0.03	0.00	0.00	0.00	0.00	0.01
Terpinolen	18.625	1088	121	0.19	0.04	0.03	0.04	0.01	0.06
Fenchone	18.727	1088	81	0.00	0.07	0.00	2.51	0.01	0.13
beta-Linalool	18.822	1101	93	0.02	0.04	0.01	0.01	0.02	0.12
1-Terpinen-4-ol	21.444	1180	71	0.11	0.01	0.00	0.01	0.01	0.03
Estragole	21.938	1196	148	0.00	0.06	0.00	1.10	0.00	0.04
Anethole	24.042	1289	148	0.00	0.02	0.00	0.33	0.03	0.02
Copaene	26.131	1394	119	0.01	0.02	0.01	0.01	0.00	0.03
beta-Caryophyllene	27.066	1444	133	0.10	0.16	0.02	0.07	0.02	0.27
alpha-Caryophyllene	27.709	1477	93	0.01	0.02	0.00	0.01	0.00	0.03
*Hydrocarbons*				2.76	4.70	5.72	7.26	2.02	11.91
*Miscellaneous*				0.22	0.09	0.08	0.32	0.38	0.05
*Unidentified*				2.16	2.71	1.14	0.74	6.02	2.17
Total chromatography peak area (×10^6^)				106	48	83	118	69	56

Products: Sse (Saucisse), Nos (Nostrano), Fel (Felino), Tos (Toscano), Sau (Saucisson) and Nap (Napoli). Peaks were identified by comparisons with mass spectra and retention index in the library of the National Institute of Standards and Technology (NIST). Tentative identification by comparison of mass spectrum with NIST library spectrum (over 85%).

## Data Availability

Not applicable.

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
