# Peer review of "Objective Measurements Associated with the Preferred Eating Qualities of Fermented Salamis"

_foods, 2021, doi:10.3390/foods10092003_

Round 1

Reviewer 1 Report

Did you mean row followed by different letters are significantly different by Tukey’s test (p < 0.05)?

296            1)Aw, water activity; Standard error of the mean. Means in the same column fol

297             lowed by different letters are significantly different by Tukey’s test (p < 0.05).

298             Products: Sauc (Saucisse), Nos (Nostrano), Fel (Felino), Tos (Toscano), Sau

 299           (Saucisson) and Nap (Napoli)

Author Response

Thank you for the opportunity to revise our manuscript. We appreciate the careful review and constructive suggestions. It is our belief that the manuscript is substantially improved after making the suggested edits. Following this letter are the reviewer comments with our responses including how and where the text was revised. Changes made in the manuscript are marked using track changes. We hope that the revision meets both of the reviewer’s and your approval.

  • Did you mean row followed by different letters are significantly different by Tukey’s test (p < 0.05)?

Answer: Thanks for the correction. The means in the same row followed by different letters.

Reviewer 2 Report

The paper needs revisions as indicated below.

  • In tables add the number of analized samples.
  • Could you change the acronym of Saucisse (Sauc) to something else? For example, Sse. It creates some confusion with Sau.
  • Line 91. How many different batches were used in the experimental scheme?
  • Line 107. Only 10 consumers? What is their distribution?
  • Lines 358-368. Have you considered the size of the solid fat particles? Small particles increase the contact surface with the sensory part of the mouth.
  • Table 5 is not clear. Line 421. “… volatile esters (14.7%)” ????
  • Lines 483 – I can’t see green colour but yellow one.

Author Response

Thank you for the opportunity to revise our manuscript. We appreciate the careful review and constructive suggestions. It is our belief that the manuscript is substantially improved after making the suggested edits. Following this letter are the reviewer comments with our responses including how and where the text was revised. Changes made in the manuscript are marked using track changes. We hope that the revision meets both of the reviewer’s and your approval.

  • In tables add the number of analized samples.

Reply. The number of replicates has been added in the tables. To make it clear, "For all analyses except volatile compounds, 6 replicates (n=6) of each sample were assessed. For the volatile compounds, four replicates (n=4) of each sample were assessed.

  • Could you change the acronym of Saucisse (Sauc) to something else? For example, Sse. It creates some confusion with Sau.

Reply. Following your suggestion, the Sauc changed to Sse.

  • Line 91. How many different batches were used in the experimental scheme?

Reply. Line 269-272: "For all analyses except volatile compounds, 6 replicates (n=6) of each sample were assessed. For the volatile compounds, four replicates (n=4) of each sample were assessed. 

  • Line 107. Only 10 consumers? What is their distribution?

 Reply. Since consumer test often comprises a large number of participants, the suitability of using only ten participants in the consumer tests might be questionable. However, a cost-effective alternative approach on a large consumer group test has been studied by Svensson (2012) found that tendencies in liking and preference between the products have been seen despite the low number of participants (ten participants), and significant differences in liking and preference was observed. Therefore, the small consumer group comprising of 10 professional consumers (Occupation: accountants, engineers and teachers; Gender: 50:50 male and female; Age: 20–40 years old; Consumption habits: fermented meat products at least once per fortnight) was performed rather than performing a large-scale random consumer test.

Reference: Svensson, Linn, 2012. Design and performance of Small scale sensory consumer tests. Second cycle, A2E. Uppsala: SLU, Dept. of Food Science

  • Lines 358-368. Have you considered the size of the solid fat particles?

Reply. We agree with your comment that the size of the solid fat particles may affects the result. The aim of this measure is to investigate fat release at in-vitro oral processing. In order to mimic mastication of oral food processing, the samples were minced finely by knife to have homogeneity of particles. Thus, the particle size effect is likely to be reduced in this study.

  • Table 5 is not clear. Line 421. “… volatile esters (14.7%)” ????

Reply. Thanks for the observation. The value has been corrected (14.7% -> 6.59%).

  • Lines 483 – I can’t see green colour but yellow one.

Reply. It has been corrected that The red colour indicates major abundance, while the yellow colour indicates minor abundance.

Reviewer 3 Report

The topic of the submitted manuscript, entitled “Objective measurements associated with the preferred eating qualities of fermented salamis” is relevant to be published in Foods.

The methods and conclusion part, should be corrected. The introduction is not clearly described. This work has a potential interest and fits with the scope of the journal. However, authors should make some changes to improve its quality.

Detailed comments:

Introduction

L.40 : „ enhancer[2].” should be replaced with: „enhancer     [2].”

L.35: “ [1] .” – should be removed the space.

L.35-37:„Post fermentation, the steps of drying and ripening are employed to concentrate flavours, decrease water activity and improve microbiological stability.” - There is lack of literature.

L.49-51: Regional factors such as ……cater to varying preferences.” - There is lack of literature.

Methods

2.4. Colour 

The color of the salami is not uniform. So why not crushed and mixed study samples?

2.5.Texture: Details of the analysis are needed.

The stacks of salami were pressed and vacuum packaged  - It should be added parameter of vacuum (pressure, producer etc.,). What kind of test was done (TPA?)? What number of probe was used? It should be used “SI” system of unite (L.136 and L.142. [mm] or [cm]).

2.6. Fat released

 “Release of moisture and fat” - should be described separately. Please add more details of methods and apparats

L.159 - Please add more details of centrifuges (e.g.manufacturer, type)

2.9. Volatile compounds

L.196-199: „…RI in the library of the National Institute of Standards and Technology (NIST) and by matching their RI elsewhere in the literature.” - What National Institute of Standards and Technology (NIST) is? It is not clear, about what literature authors mean?

2.10. Kokumi taste receptor response

Please provide more details, expand the description of the method

2.11. Statistical analysis  - There is no information about the correlation study.

Results

„Springiness” - what is the definition and unit?  Moreover, there is no discussion in the text.

L.355-357: „Fat release was not correlated to 355 the samples’ total proximate fat content (r = 0.24) but was related to their fatty acid composition (see below).” – It is not clear where this results are located?

L.360 “Likewise, Dreher, König, Herrmann, Terjung, Gibis and Weiss [17]” should be replaced with: Likewise et al. [17]”

Table 3., Parameter b*: „24 1” - what it means:

Table 5 is not clear everywhere.

Table 4: “SEMa” – what those it means  Should be explained under the table.

L. 390-391: "UFA but not SFA content of the salamis correlated with their fat released at 38 °C (r = 0.73)" - It is not clear where this results are located?

Conclusion it is too long and should be shortened.

L567- 571: There are information typicaly for introduction and should be removed.

L.591-593:” As a next step,….” – these information is not from this work, and should be deleted

Author Response

Thank you for the opportunity to revise our manuscript. We appreciate the careful review and constructive suggestions. It is our belief that the manuscript is substantially improved after making the suggested edits. Following this letter are the reviewer comments with our responses including how and where the text was revised. Changes made in the manuscript are marked using track changes. We hope that the revision meets both of the reviewer’s and your approval.

Introduction

  • 40 : „ enhancer[2].” should be replaced with: „enhancer     [2].”
  • 35: “ [1] .” – should be removed the space.
  • 35-37:„Post fermentation, the steps of drying and ripening are employed to concentrate flavours, decrease water activity and improve microbiological stability.”
  • 49-51: Regional factors such as ……cater to varying preferences.” - There is lack of literature

Reply. We agree with your comments that there were grammatical errors and a lack of literatures. Following your comments, the section has been revised.

Methods

  • 4. Colour :The color of the salami is not uniform. So why not crushed and mixed study samples?

Reply. We agree with your comment that the colour of the salami is not uniform. However, the instrumental colour meter was used to confirm whether the machine can detect colour attributes without the sample pre-preparation as did for consumer test.

  • 5.Texture: Details of the analysis are needed. The stacks of salami were pressed and vacuum packaged - It should be added parameter of vacuum (pressure, producer etc.,). What kind of test was done (TPA?)?
    What number of probe was used? It should be used “SI” system of unite (L.136 and L.142. [mm] or [cm])

Reply. We agree with your comments that some details were missing. Following as your comments, we have added all the details (definition of each parameter including unit, Information on vacuum package, probe number and TPA).

  • 6. Fat released: “Release of moisture and fat” - should be described separately. Please add more details of methods and apparats. L.159 - Please add more details of centrifuges (e.g.manufacturer, type)

Reply. Following as your comments, the information has been added.

  • 9. Volatile compounds: L.196-199: „…RI in the library of the National Institute of Standards and Technology (NIST) and by matching their RI elsewhere in the literature.” - What National Institute of Standards and Technology (NIST) is?

Reply. NIST is the standard reference databases for analytical chemistry that provides a comprehensive set of easy-to-use databases and online systems that help the analytical chemist identify unknown materials and obtain physical, chemical, and spectroscopic data about known substances. The other literatures were used to verify RI of certain compounds found in NIST database due to relatively less similarity of them. However, the expression was removed to avoid the confusion.

  • 10. Kokumi taste receptor response: Please provide more details, expand the description of the method.

Reply. Following your comment, the details on the method has been added (L 217-237).

  • 11. Statistical analysis - There is no information about the correlation study.

Reply. Line 282-283: Pearson correlation coefficients were performed between fat release and fat content or fatty acid composition.

Results

  • “Springiness” - what is the definition and unit?  Moreover, there is no discussion in the text.

Reply. The definition of springiness has been added that the height that the food recovers during the time that elapses between the end of the first bite and the start of the second bite in Line 145-147 (Material and methods). The results of other texture properties (springiness and cohesiveness) have been described in Line 379-380. However, the results were not discussed since it showed the lack of linkage between consumer description and other texture properties.

  • 355-357: Fat release was not correlated to 355 the samples’ total proximate fat content (r = 0.24) but was related to their fatty acid composition (see below).” – It is not clear where this results are located?

Reply. We apologize for the confusion. The sentence “but was related to their fatty acid composition (see below)” has been moved to section 3.6 to describe the relationship between fat release and fatty acid composition.

  • 360 “Likewise, Dreher, König, Herrmann, Terjung, Gibis and Weiss [17]” should be replaced with: Likewise et al. [17]”

Reply.  In the sentence, “Likewise” is adverb meaning “in the same way”. In order to avoid confusion, the sentence has been restructured the sentence into “Likewise, a positive contribution of melted fat to mouthfeel was reported, whereas incompletely melted fat in the mouth provides undesirable feeling [19].”

  • Table 3., Parameter b*: „24 1” - what it means:

Reply. Thanks for your observation. We corrected it: 24 1 ->24.1a.

  • Table 5 is not clear everywhere.

Reply. As per reviewer comment, to make it clear, additional information has been added in the legend of table 5 (Page 6, line 574-577; red colour highlighted).

  • Table 4: “SEMa” – what those it means?  Should be explained under the table.

Reply. The definition of SEM has been added “Standard error of the mean” under the table.

  • 390-391: "UFA but not SFA content of the salamis correlated with their fat released at 38 °C (r = 0.73)" - It is not clear where this results are located?

 Reply. Results of fat released under oral processing simulation and fatty acid composition were shown in Table 4. Pearson correlation coefficient between fat release and fatty acid composition was performed (It has been described in Statistical analysis section). Therefore, the sentence has been revised that Line 423-425: As shown in Table 4, UFA content of the salamis positively correlated with their fat released at 38 °C (r = 0.73), whereas the SFA of the salamis negatively correlated (r = -0.71).

  • L567- 571: There are information typically for introduction and should be removed.
  • 591-593:” As a next step,….” – these information is not from this work, and should be deleted

Reply. We agreed with your comments that the conclusion was not concise. Following your comments, the conclusion has been shortened and concise.